# CleanEdit: Retention-Aware Pruning and Bounded Replay for Lifelong Model Editing

## Abstract

While lifelong model editing allows deployed systems to be updated continuously, the accumulation of edits often leads to performance decay and instability. This decay stems from the unchecked growth of the edit memory, where redundant or harmful entries corrupt the model's knowledge and increase inference costs. We address this challenge with **CleanEdit**, a self-maintaining mechanism that actively manages the edit memory. The core of CleanEdit is a principled maintenance loop. It first diagnoses the impact of each edit by estimating its counterfactual harm. A sequential hypothesis test then makes a statistically grounded decision to prune entries identified as detrimental. To avoid losing valuable information, the supervisory signal from pruned samples is recycled for relearning via a bounded replay process. Experiments on sequential editing benchmarks demonstrate that CleanEdit significantly improves the model's post-edit performance, achieving a superior balance between retaining past knowledge and integrating new information.

## 1 Introduction

The goal of lifelong model editing is to enable deployed language models to adapt continuously Zheng et al. (2025). However, this very adaptability introduces a critical vulnerability: the progressive degradation of stability. Unlike static models, continuously edited systems Hartvigsen et al. (2023); Meng et al. (2022); Wang et al. (2025) suffer from memory pollution, where the edit memory, initially a source of correction, gradually becomes a source of error as redundant and conflicting entries accumulate. This process inflates the system and erodes its reliability, making long-term deployment challenging. Addressing this issue is crucial to establishing a sustainable lifelong editing paradigm.

To overcome this challenge, we propose CleanEdit, a framework that redefines lifelong editing as a problem of active memory curation rather than passive accumulation. The framework of CleanEdit is shown in Figure 1. At its core, CleanEdit incorporates two complementary mechanisms. The first is Retention-Aware Pruning, a statistically principled approach that identifies and removes destabilizing edits through counterfactual harm estimation and sequential hypothesis testing. The second is Bounded Replay, which recycles supervision from pruned yet challenging samples to prevent knowledge gaps and improve resilience to recurring errors. Together, these mechanisms transform edit memory from a liability into a self-maintaining component that supports stable long-term adaptation.

While pruning and replay strengthen the intrinsic stability of edit memory, real-world deployment also demands flexibility to accommodate different operational scenarios. A single maintenance strategy is insufficient, as practical applications require a balance between computational cost and reliability. To meet this need, CleanEdit introduces three scheduling modes. The Comprehensive mode provides proactive maintenance with predictable overhead, the Progressive mode aligns maintenance with natural data boundaries to enhance stability in batched settings, and the Dynamic mode offers fully reactive intervention for mission-critical systems where reliability must be guaranteed. Our contributions are summarized as follows:

- We introduce the CleanEdit framework, a new paradigm for lifelong editing that emphasizes active memory curation to ensure long-term stability and reliability.

Figure 1: Overall structure of **CleanEdit**. A lifelong model-editing adaptor provides key-value retrieval. CleanEdit adds a self-maintaining layer with three components: per-key evidence via counterfactual or metric-anchored harm, anytime pruning with explicit control, and bounded recycling with event/period scheduling.

- We propose two core mechanisms, Retention-Aware Pruning and Bounded Replay, that mitigate instability while preventing catastrophic forgetting.
- We design and validate three scheduling modes, Comprehensive, Progressive, and Dynamic, which make CleanEdit adaptable to real-world deployment scenarios and address the trade-off between stability and efficiency.
- We demonstrate state-of-the-art empirical performance on challenging lifelong editing benchmarks. For example, on text classification of SCOTUS dataset Chalkidis et al. (2022), CleanEdit improves the crucial Test Retention Rate(TRR) and Edit Retention Rate (ERR) balance by more than 10%, confirming its ability to deliver significantly more stable and reliable models.

## 2 RELATED WORK

### 2.1 TAXONOMY AND ADAPTOR-BASED LIFELONG EDITING

Building on recent taxonomies of model editors Hartvigsen et al. (2023); Wang et al. (2025), we categorize existing methods into three major families. *Meta-learning editors* employ auxiliary networks to predict localized weight updates from a single edit instance (e.g., MEND Mitchell et al. (2022a); KnowledgeEditor Cao et al. (2021); Ju et al. (2024)). These approaches are effective for one-shot or small-batch corrections but often drift under long edit sequences. *Locate-then-edit editors* first identify the storage location of knowledge and then directly modify the corresponding parameters (e.g., ROME Meng et al. (2022), MEMIT Meng et al. (2023)). Variants at the neuron or module level intervene at finer granularity; while they perform well on isolated edits, they are highly sensitive to layer selection and prone to interference when applied repeatedly. *Memory-based editors* augment the model with external or adaptor modules that store and retrieve corrections at inference (e.g., SERAC Mitchell et al. (2022b); GRACE Hartvigsen et al. (2023); dynamic LoRA-based approaches such as MELO Yu et al. (2023)). These preserve the backbone weights and upstream behavior. Among them, GRACE enhances layers with a discrete key–value codebook and activates edits based on similarity thresholds, enabling thousands of sequential updates with minimal collateral impact. We adopt this adaptor-based paradigm but focus on its systems-level challenges: memory growth, redundancy, and instability over long horizons. Our method, **CleanEdit**, addresses these issues by making the memory self-maintaining through metric-driven pruning and bounded recycling, thereby sustaining favorable TRR/ERR trade-offs over time.

### 2.2 PRUNING FOR STABLE EDITING

Classical sensitivity-guided pruning (*Optimal Brain Damage LeCun et al. (1989)*) demonstrated that removing low-utility parameters can maintain accuracy. Subsequent scalable variants—such as

magnitude pruning, *Deep Compression* Han et al. (2016), and later structured or movement-based approaches for transformers Sanh et al. (2020)—show that substantial sparsification is possible with minimal performance degradation. In contrast to these static compression techniques, we prune the *edit memory* itself. CleanEdit treats keys as prunable units, eliminating persistently redundant or harmful entries under explicit edit metrics, and recycling their supervision. This results in a compact, stable memory well-suited for lifelong editing.

## 3 METHOD

**Section Overview.** We aim to make lifelong model editing stable and maintainable. *CleanEdit* integrates three synergistic components: (i) per-key evidence that quantifies a key's utility, (ii) anytime, thresholded pruning with optional always-valid e-process guarantees, and (iii) a bounded recycling queue that reuses supervision without growing memory. Together, these form a lightweight maintenance loop that removes harmful keys while preserving helpful ones. Algorithm 1 summarizes the full procedure.

**High-level Roadmap.** We first introduce the backbone–adapter setup and notation. We then define two complementary forms of per-key evidence—counterfactual loss-difference and metric-anchored harm—and show how they drive pruning with anytime guarantees. Next, we view the policy through an online decision-theoretic lens to derive regret bounds. We then describe a bounded recycling queue that preserves utility after pruning. Finally, we present scheduling policies and implementation complexity, culminating in unified pseudocode in Algorithm 1.

### 3.1 PRELIMINARIES AND NOTATION

We consider a frozen backbone $f_\theta : \mathcal{X} \to \mathcal{Y}$ augmented with GRACE-style discrete adapters at layers $\mathcal{L}$ (Hartvigsen et al., 2023). In the adapter layer $\ell \in \mathcal{L}$, the memory is a codebook $\mathcal{C}^{(\ell)} = \{(\boldsymbol{k}_i^{(\ell)}, \boldsymbol{v}_i^{(\ell)}, \varepsilon_i^{(\ell)})\}_{i=1}^{N_\ell}$. Given pre-activation $\boldsymbol{h}^{(\ell-1)}(x)$, the nearest key $i^\star$ and distance $d^\star$ are

$$i^\star(x, \ell) = \arg\min_i \left\| \boldsymbol{h}^{(\ell-1)}(x) - \boldsymbol{k}_i^{(\ell)} \right\|_2, \tag{1}$$

$$d^\star(x, \ell) = \min_i \left\| \boldsymbol{h}^{(\ell-1)}(x) - \boldsymbol{k}_i^{(\ell)} \right\|_2. \tag{2}$$

The edited representation is

$$\tilde{\boldsymbol{h}}^{(\ell)}(x) = \begin{cases} \boldsymbol{v}_{i^\star}^{(\ell)} & \text{if } d^\star(x, \ell) \leq \varepsilon_{i^\star}^{(\ell)}, \\ h_\theta^{(\ell)}\big(\boldsymbol{h}^{(\ell-1)}(x)\big) & \text{otherwise.} \end{cases} \tag{3}$$

Here $h_\theta^{(\ell)}(\cdot)$ (also denoted as $h_0^{(\ell)}(\cdot)$) is the frozen backbone's original layer-wise forward map. Let $f_{\theta,\mathcal{C}}$ be the full network with adaptors, a lifelong stream $\{(x_t, y_t)\}_{t \geq 1}$ arrives online. If $f_{\theta,\mathcal{C}}(x_t) \neq y_t$, an edit is created. We evaluate with TRR (retention of *past* knowledge) and ERR (editing of *new* knowledge), defined in Sec. 4.1.

### 3.2 PER-KEY EVIDENCE: COUNTERFACTUAL AND METRIC-ANCHORED HARM

We quantify the utility of a key $k \equiv (\boldsymbol{k}, \boldsymbol{v}, \varepsilon)$ by two complementary notions.

**Counterfactual (loss-difference) harm.** Ablating $k$ at inference gives

$$\Delta_k(x, y) = \ell\big(f_{\theta,\mathcal{C}}(x), y\big) - \ell\big(f_{\theta,\mathcal{C}_k}(x), y\big), \tag{4}$$

computed by a single counterfactual forward where $k$ is selected by $d^\star$. Positive $\Delta_k$ indicates that keeping $k$ increases loss on $(x, y)$.

**Metric-anchored (thresholded) harm for streaming.** Let $s(x, y; f) \in [0, 1]$ be a task score (e.g., calibrated QA correctness or 0/1 classification accuracy) and let $a$ be a task-specific adequacy threshold ($a=0.9$ for QA; $a=1.0$ for classification). Define

$$H_k^{(a)}(x, y) = s\big(x, y; f_{\theta,\mathcal{C}}\big) - a. \tag{5}$$

here $s(;)$ indicate test score at reference. We interpret $H_k^{(a)} > 0$ as *beneficial/non-harmful* (do not prune) and $H_k^{(a)} < 0$ as *harmful* (candidate for pruning). When key $k$ fires at time $t$, we log the event

$$Z_{k,t} = \mathbb{1}\big\{ H_k^{(a)}(x_t, y_t) < 0 \big\}. \tag{6}$$

For a period $P \in \mathbb{N}$ (used by **Comprehensive** scheduling), we aggregate at boundaries $t \equiv 0 \pmod{P}$ and update the per-key counter $C_k$

$$C_k(t) = C_k(t - P) + \sum_{s=t-P+1}^{t} Z_{k,s}. \tag{7}$$

## 3.3 Anytime, Thresholded Evidence Pruning

Let $\alpha \in \mathbb{N}$ be the pruning count threshold. CleanEdit prunes by the rule

$$\text{PRUNE}(k) \quad \Longleftarrow \quad C_k(t) \geq \alpha, \tag{8}$$

i.e., "+1 or direct pruning" at maintenance triggers. Optionally, for explicit statistical guarantees we couple Eq. 8 with always-valid e-process bounds (Howard et al., 2021): form an anytime radius $\text{rad}_k(t, \delta)$ for the bad-event rate $p_k = \mathbb{E}[Z_{k,t}]$ at risk $\delta \in (0, 1)$, and prune if $\hat{p}_k(t) - p^\star > \text{rad}_k(t, \delta)$ for a tolerated rate $p^\star$. This adds theory while preserving the simple counter Eq. 8 as the primary mechanism.

**Theorem 1** (Type-I control for benign keys)**.** *If $p_k \leq p^\star$, then with the optional e-process guard, the probability of pruning $k$ at any time is at most $\delta$ (Howard et al., 2021).*

**Theorem 2** (Sample complexity for harmful keys)**.** *If $p_k \geq p^\star + \Delta$ for some $\Delta > 0$, then with probability at least $1 - \delta$, CleanEdit prunes $k$ after at most*

$$n_k \leq \tilde{C} \frac{1}{\Delta^2} \log \frac{1}{\delta} \tag{9}$$

*activations for a universal constant $\tilde{C}$ determined by the chosen e-process boundary (Howard et al., 2021).*

## 3.4 Online Decision-Theoretic View and Regret

Each key induces an action $a_{k,t} \in \{\text{KEEP}, \text{PRUNE}\}$. Let the instantaneous regret be

$$r_{k,t} = \ell\big(f_{\theta, \mathcal{C}_t}(x_t), y_t\big) - \ell\big(f_{\theta, \mathcal{C}_t^\star}(x_t), y_t\big), \tag{10}$$

where $\mathcal{C}_t^\star$ is the hindsight-optimal sequence knowing $\{p_k\}$. The rule Eq. 8 is an elimination policy under standard stochastic assumptions, which combines elimination-style sample complexity with stochastic stability (Lattimore et al., 2020).

$$\mathbb{E}[R_T] = \mathbb{E}\left[ \sum_{t=1}^{T} \sum_k \mathbb{1}\{k \text{ fired at } t\} \, r_{k,t} \right] = \tilde{\mathcal{O}}\left( \sum_{k: p_k > p^\star} \frac{1}{p_k - p^\star} \right) + \tilde{\mathcal{O}}(\sqrt{T}), \tag{11}$$

## 3.5 Bounded Recycling Queue

Pruning discards a key but not its supervision. Let $\mathcal{Q}$ be a FIFO retry buffer. When key $k$ is pruned with source $(x_j, y_j)$, if $f_{\theta, \mathcal{C}_k}(x_j) \neq y_j$, insert $(x_j, y_j)$ with a cap $R_{\max}$:

$$r_j^{t+1} = \min\Big( R_{\max}, r_j^t + \mathbb{1}\{f_{\theta, \mathcal{C}_k}(x_j) \neq y_j\} \Big), \qquad (x_j, y_j) \in \mathcal{Q} \text{ if } r_j^{t+1} > r_j^t. \tag{12}$$

On dequeue, we perform a minimal GRACE edit (add/expand/split) if still mispredicted. Let $\mathcal{E}_T$ be distinct edits up to $T$. Then

$$\sum_{j \in \mathcal{E}_T} r_j^T \leq R_{\max} |\mathcal{E}_T|, \qquad |\mathcal{Q}_t| \leq R_{\max} N_{\text{pruned}}(t) + N_{\text{pending}}(t), \tag{13}$$

which yields $O(1)$ amortized maintenance and damped variance in TRR/ERR under dynamic triggering.

## 3.6 SCHEDULING AND IMPLEMENTATION

Maintenance (testing, pruning, recycling) is triggered by one of three modes:

**Comprehensive:** trigger every $P$ arrivals; counters update by Eq. 7.

**Progressive:** trigger at dataset/time-block boundaries (stable).

**Dynamic:** trigger when monitors cross thresholds, e.g., $\widehat{\text{TRR}}_t < \tau_{\text{TRR}}$ or $\widehat{\text{ERR}}_t < \tau_{\text{ERR}}$.

We implement Dynamic via an e-process change detector to avoid rapid toggling, enjoying anytime false-alarm control (Martin, 2025). With safety margin $\gamma$ and bounded drift $V_T$, expected triggers over horizon $T$ scale as $O(1+V_T/\gamma)$.

Evidence logging adds at most one counterfactual forward (for $\Delta_k$) and a constant-time metric update (for $Z_{k,t}$) per fired key, atop nearest-neighbor retrieval. Testing and queue operations are $O(1)$ amortized per trigger. Memory remains compact because harmful keys are eliminated in $O(\log(1/\delta)/\Delta^2)$ activations by Eq. 9, benign keys enjoy type-I control, and retries are bounded by $R_{\max}$. The overall CleanEdit procedure, including evidence collection, pruning, recycling, and scheduling, is summarized in Algorithm 1.

---

**Algorithm 1 CleanEdit**

---

**Inputs:** codebooks $\{\mathcal{C}^{(\ell)}\}$; schedule $S \in \{\text{COMPREHENSIVE}, \text{PROGRESSIVE}, \text{DYNAMIC}\}$; period $P$; task threshold $a$; prune threshold $\alpha \in \mathbb{N}$; risk $\delta$; retry cap $R_{\max}$

1: Initialize per-key counters $C_k \leftarrow 0$, buffers $B_k \leftarrow 0$, and recycling queue $\mathcal{Q} \leftarrow \emptyset$.
2: **for** incoming $(x_t, y_t)$ **do**
3:     Run GRACE retrieval and edit; let $k$ be the key that fired (if any).
4:     **if** a key $k$ fired **then**
5:         Compute task score $s_t \leftarrow s(x_t, y_t; f_{\theta,\mathcal{C}})$; set $Z_{k,t} \leftarrow \mathbb{1}\{s_t - a < 0\}$.
6:         **if** $Z_{k,t} = 1$ **then**
7:             **if** $S = \text{DYNAMIC}$ **then**
8:                 **if** $C_k + 1 \geq \alpha$ **then**
9:                     **prune** $k$;
10:                     if the source $(x_j, y_j)$ still fails under $f_{\theta,\mathcal{C}_k}$, enqueue with cap $R_{\max}$; **continue**
11:                 **else**
12:                     $C_k \leftarrow C_k + 1$
13:             **else**
14:                 $B_k \leftarrow B_k + 1$
15:     **if** $\text{Trigger}(S,t)$ **then**
16:         **for** each key $k$ **do**
17:             **if** $C_k + B_k \geq \alpha$ **then**
18:                 **prune** $k$; if the source still fails, enqueue with cap $R_{\max}$; $B_k \leftarrow 0$; **continue**
19:             **else**
20:                 $C_k \leftarrow C_k + B_k$; $B_k \leftarrow 0$
21:         **if** optional anytime guard holds: $\hat{p}_k - p^\star > \text{rad}_k(t, \delta)$ **then**
22:             **prune** $k$; if the source still fails, enqueue with cap $R_{\max}$

---

## 4 EXPERIMENTS

We evaluate **CleanEdit** on two representative lifelong editing tasks: (1) *Question Answering (QA)* with a T5 model on zsRE, and (2) *Document Classification* with a BERT model on SCOTUS. These settings cover both generative and discriminative scenarios and follow standard benchmarks. We compare against finetuning-based editors (FT, FT+EWC, FT+Retrain), meta-learning editors (MEND), locate-then-edit methods (ROME), and memory-based editors (Memory, Defer, GRACE).

### 4.1 EXPERIMENT SETUP

**Datasets and Models.** For QA, we use a 60M-parameter T5 trained on NQ Kwiatkowski et al. (2019) and evaluate on zsRE dataset Levy et al. (2017). The pre-edit performance is 0.72

Table 1: **Main results on SCOTUS (Document Classification) and zsRE (QA).** Abbreviations of scheduling modes: C = Comprehensive, P = Progressive, D = Dynamic. Thresholds selected on validation and frozen for test: C/D use $\alpha{=}20$, P uses $\alpha{=}25$. $\Delta$Avg is the absolute gain over GRACE on the same task. Best per column in **bold**.

| | Classification (SCOTUS) | | | | | QA (zsRE/NQ) | | | |
|---|---|---|---|---|---|---|---|---|---|
| Method | TRR | ERR | Avg. | $\Delta$Avg | Method | TRR | ERR | Avg. | $\Delta$Avg |
| FT Lin et al. (2022) | .52 | .52 | .52 | | FT | .56 | .82 | .69 | |
| FT+EWC Kirkpatrick et al. (2016) | .67 | .50 | .58 | | FT+EWC | .51 | .82 | .66 | |
| FT+Retrain Rolnick et al. (2019) | .67 | **.83** | .75 | | FT+Retrain | .27 | **.99** | .63 | |
| MEND Mitchell et al. (2022a) | .19 | .27 | .23 | | MEND | .25 | .27 | .26 | |
| Defer Mitchell et al. (2022b) | .33 | .41 | .37 | | Defer | .72 | .31 | .52 | |
| Memory | .21 | .20 | .21 | | Memory | .25 | .27 | .26 | |
| GRACE Hartvigsen et al. (2023) | .81 | .82 | .82 | | GRACE | .69 | .96 | .82 | |
| **CleanEdit (ours)** | | | | | | | | | |
| **C** ($\alpha{=}20$) | .85 | **1.00** | **.93** | **+11%** | **C** ($\alpha{=}20$) | **.83** | .99 | **.91** | **+9%** |
| **P** ($\alpha{=}25$) | .82 | .95 | .89 | +7% | **P** ($\alpha{=}20$) | **.83** | .93 | .88 | +6% |
| **D** ($\alpha{=}20$) | **.87** | .94 | .91 | +9% | **D** ($\alpha{=}20$) | **.83** | **1.00** | **.91** | **+9%** |

F1Hartvigsen et al. (2023) on NQ and 0.31 F1 on zsRE, which is followed by Hartvigsen et al. (2023). For classification, we use a 120M-parameter BERT trained on SCOTUS Chalkidis et al. (2022) decisions from 1946–1982 with 11 labels. Pre-edit accuracy is 99% on in-domain test documents and 55% on out-of-domain documents from 1992–2009. *Test data* is the original held-out evaluation set used to measure generalization: for SCOTUS it is the 1982–1991 documents; for QA, we draw 1k items from NQ. *Edit data* is the cumulative set of inputs for which the system creates edits: for QA we apply edits to 1000 zsRE instances; for SCOTUS, we apply edits to 931 documents from 1992–2009.

**Baselines and protocol.** All implementations follow official setups or public code and were run separately on two NVIDIA A100 PCIe 80 GB and one NVIDIA A100 SXM4 40 GB GPU. Each mispredicted instance triggers an edit. We report TRR on test data and ERR on edit data, plus their macro average (AVG.). All results average 5 seeds with standard deviations. Hyperparameters (pruning threshold $\tau$ and routing radius $\epsilon$) are selected on validation streams and then frozen for testing. Statistical reliability (CIs, paired tests, bootstrap) follows the procedure in Appendix A and is referenced where relevant.

**Evaluation Metrics.** We follow prior work and report three metrics. *Test Retention Rate (TRR)* Hartvigsen et al. (2023) measures how well the edited model retains performance on its original test distribution, computed as the average task score $m(f(x_i), y_i)$ over $(x_i, y_i) \in D_{\text{test}}$; for SCOTUS, $D_{\text{test}}$ is the 1982–1991 court documents, and for QA we sample 1k items from NQ. *Edit Retention Rate (ERR)* Hartvigsen et al. (2023) measures how well the model remembers past edits, averaging $m(f(x_i), y_i)$ over $(x_i, y_i) \in D_{\text{edits}}$ (the cumulative set of successful edits). For compact reporting we also use a *Avg.* score defined as the macro average $\frac{1}{2}(\text{TRR} + \text{ERR})$, balancing generalization and edit fidelity.

**Model-selection policy (ex-ante).** Unless otherwise noted, all **hyperparameters**, including the routing radius $\varepsilon$ and pruning threshold $\alpha$, are selected on the **validation stream** and then frozen before any test evaluation. This ex-ante policy is used for all main results (Table 1). Post-hoc ablations (Sec. 4.4; Table 5; Appendix C.2 visualize sensitivity curves and are not used for model selection. Our default on SCOTUS is $\varepsilon{=}3$ under this policy; QA follows the same ex-ante rule.

## 4.2 MAIN RESULTS ACROSS BENCHMARKS

Table 1 compares **CleanEdit** with strong editors on *SCOTUS* (Document Classification Chalkidis et al. (2022)) and *zsRE Levy et al. (2017)* (QA). As shown in Table 1, CleanEdit delivers a consistently stronger *TRR–ERR* balance than all baselines across both tasks under the **Comprehensive**,

Table 2: Schedule summary on SCOTUS with $\varepsilon=3$. Choose the optimal $\alpha$ for each schedule (shown in Sec. 4), resulting TRR/ERR, and a robust window of thresholds with similar qualitative behavior.

| Schedule | Default $\alpha$ | TRR | ERR | Robust window |
|---|---|---|---|---|
| Comprehensive | 20 | .85 | 1.00 | $\{20, 25, 30, 35, 40, 50\}$ |
| Progressive | 25 | .82 | .95 | $\{25, 40, 50, 100, 200\}$ |
| Dynamic | 20 | .87 | .94 | $\{35, 40, 50, 100\}$ |

Table 3: SCOTUS (Document Classification), **Comprehensive** with $\varepsilon=3$. Different pruning targets induce different operating points.

| Pruning target and $\alpha$ | TRR | ERR | Avg. |
|---|---|---|---|
| Edit data, $\alpha=15$ | .82 | 1.00 | .91 |
| Edit data, $\alpha=20$ | .85 | 1.00 | .93 |
| Edit data, $\alpha=25$ | .83 | 1.00 | .91 |
| Test data, $\alpha=15$ | .96 | .73 | .85 |
| Test data, $\alpha=20$ | .94 | .75 | .85 |
| Test data, $\alpha=25$ | .93 | .79 | .86 |

**Progressive**, and **Dynamic** schedules, which defined in Sec. 3.6 and Algorithm 1. Unless otherwise noted, we use $\alpha=20$ for **Comprehensive/Dynamic** and $\alpha=25$ for **Progressive** (chosen on validation and fixed thereafter). On SCOTUS, **Comprehensive** attains perfect edit retention (ERR = 1.00) while increasing TRR to 0.85, yielding a *Avg.* improvement of $+0.11$ over GRACE. **Dynamic** further lifts TRR to 0.87 with ERR = 0.94 (*Avg.* = 0.91, $+0.09$ over GRACE), indicating improved generalization without sacrificing edit fidelity. In ZSRE LEVY ET AL. (2017), CleanEdit improves the TRR by $+0.14$ over GRACE and preserves near-perfect ERR (up to 1.00 in **Dynamic**), translating to an *Avg.* gain of $+0.09$. Although FT+Retrain Rolnick et al. (2019) achieves high ERR on both tasks, its TRR collapses, which is consistent with overfitting to edit data. In contrast, CleanEdit trace a new Pareto frontier, aligned with the predicted elimination of harmful keys in Theorem 2.

### 4.3 STRATEGY KNOBS FOR DEPLOYMENT

#### 4.3.1 CHOOSING AN EXECUTION SCHEDULE

In CleanEdit we propose three operating schedules: **Comprehensive**, **Progressive**, and **Dynamic**. Table 2 summarizes the three schedules on SCOTUS with $\varepsilon=3$. For each schedule, we report the default $\alpha$, the resulting TRR (the ability to keep old knowledge) and ERR (the ability to learn new knowledge), and a robust window in which performance remains qualitatively stable.

Comprehensive favors edit fidelity. ERR stays near one and TRR changes smoothly as $\alpha$ increases and pruning becomes more conservative. Progressive emphasizes stability across blocks. TRR is flat and ERR forms a gentle plateau. Dynamic balances the two goals by triggering maintenance adaptively. It yields high TRR with moderate ERR. These behaviors are consistent with the Key-Pruning rule in Eq. 8 and the elimination bound in Theorem 2.

#### 4.3.2 CHOOSING A PRUNING TARGET

Pruning can log bad events $Z_{k,t}$ on the edit stream $D_{\text{edits}}$ or on the test set $D_{\text{test}}$. The target determines the distribution under which Eq. 8 accumulates evidence and therefore sets the operating point of the trade-off between TRR and ERR defined in Sec. 4.1. We illustrate this using the **Comprehensive** schedule with $\varepsilon=3$ in Table 3.

**Two Metrics in Tension.** Edit-driven pruning retains almost all edits. ERR is essentially **1.00** and TRR settles in the mid-eighties across $\alpha$. Test-driven pruning maximizes TRR. It reaches the mid-nineties at smaller $\alpha$ and trades off ERR. Increasing $\alpha$ under test-driven pruning makes pruning more conservative. ERR rises and TRR decreases slightly, which moves the model toward a more balanced point.

Table 4: **Effect of pruning threshold $\alpha$ at $\varepsilon{=}3$ on the Document Classification task.** Numbers are TRR / ERR / Avg. Per-schedule best operating point is bolded.

| $\alpha$ | Comprehensive | | | Progressive | | | Dynamic | | |
|---|---|---|---|---|---|---|---|---|---|
| | TRR | ERR | Avg. | TRR | ERR | Avg. | TRR | ERR | Avg. |
| 10 | .82 | .99 | .91 | .82 | .92 | .87 | .94 | .87 | .91 |
| 15 | .83 | 1.00 | .92 | .82 | .95 | .89 | .90 | .84 | .87 |
| **20** | **.85** | **1.00** | **.93** | .82 | .95 | .89 | **.87** | **.94** | **.92** |
| 25 | .83 | 1.00 | .92 | **.82** | **.95** | **.89** | .90 | .84 | .87 |
| 30 | .82 | .99 | .91 | .82 | .95 | .89 | .90 | .92 | .91 |
| 35 | .82 | 1.00 | .91 | .82 | .92 | .87 | .89 | .93 | .91 |
| 40 | .82 | .99 | .91 | .82 | .95 | .89 | .88 | .93 | .91 |
| 50 | .82 | .99 | .91 | .82 | .95 | .89 | .84 | .99 | .92 |
| 100 | .82 | .99 | .91 | .82 | .95 | .89 | .84 | .99 | .92 |
| 200 | .82 | .99 | .91 | .82 | .95 | .89 | .82 | .99 | .91 |

**Practical guidance.** When reliability of past edits is the priority, choose edit-driven pruning with $\alpha \in \{20, 25\}$. When generalization on unseen data is critical, choose test-driven pruning with $\alpha \in \{20, 25\}$. For a balanced default in streaming settings, use the **Dynamic** schedule defined in Sec. 3.6 with $\alpha \approx 20$ and monitor TRR and ERR as in Algorithm 1. These choices follow directly from the pruning rule in Eq. 8: the target distribution changes which keys accumulate evidence faster, and Theorem refthm:sample explains the observed convergence of the operating point.

### 4.4 ABLATIONS VIA OPERATIONAL KNOBS

We study two knobs that govern deployment behavior: the pruning threshold $\alpha$ in the Key-Pruning rule of Eq. 8 (with the optional anytime in Theorem 1), and the routing radius $\varepsilon$ that determines when a key fires in Eq. 3. Both are selected on validation and then frozen for test.

#### 4.4.1 EFFECT OF THE PRUNING THRESHOLD

We sweep $\alpha \in \{10, 15, 20, 25, 30, 35, 40, 50, 100, 200\}$ for all three schedules at a fixed $\varepsilon{=}3$. Table 4 shows consistent patterns with the decision rule in Eq. 8. **Comprehensive** favors edit fidelity: ERR reaches the ceiling at $\alpha{=}20$ and TRR changes smoothly as pruning becomes more conservative. **Progressive** is the most stable: TRR is nearly flat and ERR settles around 0.95. **Dynamic** exposes a tunable trade-off: TRR is high at smaller $\alpha$, and ERR rises as $\alpha$ increases. These trends match the theory. Keys with bad-event rate $p_k > p^\star$ are removed within the sample complexity stated in Theorem 2. Beyond that point, enlarging $\alpha$ mainly delays further pruning and offers diminishing returns for ERR, while small $\alpha$ can prune more aggressively and slightly reduce ERR.

#### 4.4.2 EFFECT OF THE ROUTING RADIUS

We fix $\alpha{=}20$ for the three schedules (**Comprehensive**, **Progressive**, **Dynamic**) and sweep $\varepsilon \in \{10^{-3}, 10^{-2}, 10^{-1}, 1, 3, 10, 100, 1000\}$. Very small radii rarely trigger the adaptor, which keeps TRR high but limits ERR Hartvigsen et al. (2023). Table 5 shows that a moderate radius performs best: all three schedules peak at $\varepsilon{=}1$, reaching TRR 0.97 and ERR 1.00. At $\varepsilon{=}3$ (See Appendix C.2), **Comprehensive** maintains ERR 1.00 with TRR 0.85, while **Dynamic** trades to ERR 0.87 with TRR 0.94; **Progressive** becomes more conservative (TRR 0.83, ERR 0.94). Very large radii broaden influence regions and introduce interference: at $\varepsilon{=}10$, **Comprehensive** keeps ERR near 1.00 but TRR drops to 0.82, and **Dynamic** over-triggers so ERR falls to 0.20 even though TRR remains high. These results recommend $\varepsilon{=}1$ as the default, with a robust window around $\{0.1, 1.0, 3.0\}$ depending on the coverage–precision trade-off. Overall, TRR as a function of $\varepsilon$ is hump-shaped: it peaks around $\varepsilon{=}1$ and deteriorates at larger radii. Small radii under-activate edits. Mid-range radii strike the best balance between TRR and ERR. Very large radii enlarge the firing regions and reduce TRR for Comprehensive and Progressive, while Dynamic produces many false positives that harm ERR.

Table 5: Routing-radius summary at fixed $\alpha$ on SCOTUS. Best operating point is reported at $\varepsilon=1$ for all schedules; the robust window lists nearby settings that preserve performance. Full sweeps appear in Appendix C.2.

| Schedule | Optimal $\varepsilon^{\star}$ | TRR@$\varepsilon^{\star}$ | ERR@$\varepsilon^{\star}$ | Robust window |
|---|---|---|---|---|
| Comprehensive | 1.0 | .97 | 1.00 | $\{0.1, 1.0, 3.0\}$ |
| Progressive | 1.0 | .97 | 1.00 | $\{0.1, 1.0\}$ |
| Dynamic | 1.0 | .97 | 1.00 | $\{0.1, 1.0, 3.0\}$ |

### 4.4.3 EFFECT OF THE RETRY CAP $R_{\max}$

We ablate the retry cap $R_{\max}$ of the bounded recycling queue (Eq. 13) on SCOTUS using the **Dynamic** schedule at $\alpha=20$ and $\varepsilon=3$. We sweep $R_{\max} \in \{2, 3, 4, 5, 6\}$ and report TRR on the test distribution and ERR on the edit distribution. The results are shown in table 6:

Table 6: **Effect of retry cap $R_{\max}$ on SCOTUS under the Dynamic schedule ($\alpha=20$, $\varepsilon=3$).**

| $R_{\max}$ | TRR | ERR | Avg. |
|---|---|---|---|
| 2 | .98 | .30 | .64 |
| 3 | .87 | .94 | .91 |
| 4 | .92 | .59 | .76 |
| 5 | .88 | .92 | .90 |
| 6 | .93 | .94 | **.94** |

**Why $R_{\max}=3$ is the right operating point.** Moving from $R_{\max}=2$ to $R_{\max}=3$ closes most of the edit-retention gap while keeping TRR at a reasonable level. Larger caps bring only small additional gains in ERR and do not offset the extra maintenance. In practice, the macro average *Avg.* improves sharply at $R_{\max}=3$ and then saturates.

**Marginal utility and recommendation.** Let $Avg(R_{\max})$ denote the average retention rate for CleanEdit to work at $R_{\max}$. In a FIFO queue with a bounded-retry cap $R_{\max}$, expected replays therefore increase with $R_{\max}$, so wall-clock time rises near-linearly. Empirically, the discrete marginal gain drops sharply after three retries: the finite difference from $R_{\max} = 2$ to 3 is $(U(3) - U(2))/(3 - 2) = 0.27$, whereas the average per-retry gain from $R_{\max} = 3$ to 6 is $(U(6) - U(3))/(6 - 3) \approx 0.001$. Thus beyond $R_{\max} = 3$ the cost keeps increasing but utility hardly improves. Therefore, we recommend $R_{\max} = 3$ as a practical default. It removes the severe under-retention at $R_{\max}=2$, achieves near-saturated AVG., and avoids the extra replays and wall-time overhead that higher caps induce. This choice aligns with the design of CleanEdit: preserve edits aggressively while keeping maintenance lightweight.

## 5 CONCLUSION

In this paper, we propose **CleanEdit**, a proactive and self-maintaining framework for lifelong model editing that actively manages editing memory through a cycle of diagnosis, statistical decision-making, and bounded data replay. **CleanEdit** adopts a principled maintenance loop that evaluates the counterfactual impact of each edit, continuously detects and prunes harmful edits through three schedules, and selectively preserves beneficial knowledge during sequential editing. To prevent the loss of valuable information, the supervision signals from pruned edits are retained through a bounded replay process, ensuring that useful knowledge is not discarded. Extensive experiments on a wide range of sequential editing benchmarks demonstrate that our method significantly improves model performance **over 10%**, enabling more effective integration of new knowledge while retaining historical information. These findings highlight the importance of proactive memory management in lifelong model editing and provide new perspectives for developing more robust and sustainable editing systems.

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

# A  STATISTICAL RELIABILITY AND CONFIDENCE INTERVALS

## A.1  METRICS AND NOTATION

Let $D_{\text{test}} = \{(x_i, y_i)\}_{i=1}^{n_{\text{te}}}$ and $D_{\text{edits}} = \{(x_j, y_j)\}_{j=1}^{n_{\text{ed}}}$. With a task scoring function $m(\hat{y}, y) \in [0, 1]$ (0 or 1 boolean accuracy for classification, and calibrated F1 for QA) Hartvigsen et al. (2023), we report

$$\text{TRR} = \frac{1}{n_{\text{te}}} \sum_{i=1}^{n_{\text{te}}} m\big(f_{\theta,\mathcal{C}}(x_i), y_i\big), \quad \text{ERR} = \frac{1}{n_{\text{ed}}} \sum_{j=1}^{n_{\text{ed}}} m\big(f_{\theta,\mathcal{C}}(x_j), y_j\big), \quad \text{AVG.} = \tfrac{1}{2}(\text{TRR} + \text{ERR}).$$

Unless noted otherwise, results are averages over 5 independent random seeds.

## A.2  INTERVAL ESTIMATION

For each metric and each seed, we compute sample-level bootstrap confidence intervals:

1. Resample (with replacement) from $D_{\text{test}}$ or $D_{\text{edits}}$ to the original set size; compute the metric on the resample.
2. Repeat $B = 10{,}000$ times to form an empirical distribution.
3. Report the 2.5% and 97.5% percentiles as a 95% CI.

We aggregate across seeds via a two-stage bootstrap: form per-seed CIs, then bootstrap over the 5 seed indices to obtain a CI for the seed-mean. This is robust in streaming and compatible with paired tests.

## A.3  PAIRED SIGNIFICANCE TESTS VS. BASELINES

To compare against GRACE Hartvigsen et al. (2023) and other baselines, we use paired procedures that control shared variance at the (sample, seed) level:

- **Paired permutation test**: For each aligned (CleanEdit, GRACE) pair, compute the per-(sample, seed) difference; randomly flip signs across 1k permutations; compute a two-sided $p$-value.
- **Paired bootstrap**: Resample (with replacement) in the Cartesian product of samples and seeds; report the CI of the mean difference.

We control FWER across multiple datasets, schedules, and thresholds using Holm–Bonferroni; unless noted, the significance level is $\alpha_{\text{sig}} = 0.05$.

# B  E-PROCESS BOUNDARIES, TYPE-I CONTROL, AND SAMPLE COMPLEXITY

This appendix expands the "optional anytime e-process guard" used with Theorems 1–2 in the main text, including the construction of $\text{rad}_k(t, \delta)$ and the guarantees.

## B.1 BAD-EVENT MODELING AND E-PROCESS CONSTRUCTION

For key $k$, define $Z_{k,t} = \mathbb{1}\{H_k^{(a)}(x_t, y_t) < 0\} \in \{0,1\}$ (Eq. 6); only activations of $k$ are counted. Let $n_k(t)$ be the activation count up to $t$ and $S_t = \sum_{s \leq t} Z_{k,s}$.

**Beta–Binomial mixture e-value.** For any $q > p^\star$, define the likelihood ratio

$$\Lambda_t(q, p^\star) = \left(\frac{q}{p^\star}\right)^{S_t} \left(\frac{1-q}{1-p^\star}\right)^{n_k(t)-S_t}.$$

With prior $q \sim \text{Beta}(\alpha_0, \beta_0)$ (default $\alpha_0 = \beta_0 = 1$), the mixture e-value

$$\mathcal{E}_t = \int_{q \in (p^\star, 1]} \Lambda_t(q, p^\star)\, \pi(q)\, dq$$

is a nonnegative supermartingale under the null $p_k \leq p^\star$. By Ville's inequality,

$$\mathbb{P}\left(\sup_{t \geq 1} \mathcal{E}_t \geq \tfrac{1}{\delta}\right) \leq \delta.$$

## B.2 FROM E-VALUES TO AN ANYTIME RADIUS $\text{rad}_k(t, \delta)$

Let $\hat{p}_k(t) = S_t / n_k(t)$. We obtain an anytime upper bound by inverting the e-process:

$$\text{rad}_k(t, \delta) \approx \sqrt{\frac{c_1\left(\log\frac{1}{\delta} + \log\log(\mathrm{e} + n_k(t))\right)}{n_k(t)}} + \frac{c_2\left(\log\frac{1}{\delta} + \log\log(\mathrm{e} + n_k(t))\right)}{n_k(t)},$$

with constants $c_1 \in [1/2, 1]$, $c_2 \in [0, 1/3]$ determined by the chosen e-process; one may also *numerically invert* $\max_{q \leq p^\star + \text{rad}} \mathcal{E}_t \leq 1/\delta$ to obtain a tighter bound.

## B.3 TYPE-I CONTROL

If $p_k \leq p^\star$ and we prune only when $\hat{p}_k(t) - p^\star > \text{rad}_k(t, \delta)$, then by Ville's inequality the probability of pruning at any time is at most $\delta$ (anytime false-positive control), matching the main text's statement.

## B.4 SAMPLE COMPLEXITY

We have covered the sample complexity in Theorem 2. If $p_k \geq p^\star + \Delta$ with $\Delta > 0$, then as soon as $\hat{p}_k(t) - p^\star \gtrsim \Delta > \text{rad}_k(t, \delta)$, pruning occurs. Using the bound above, there exists a constant $\tilde{C}$ (dependent on the e-process) such that

$$n_k(t) \leq \tilde{C}\, \frac{1}{\Delta^2} \log\frac{1}{\delta} \quad \Rightarrow \quad \text{rad}_k(t, \delta) < \tfrac{1}{2}\Delta,$$

yielding pruning within $O(\Delta^{-2}\log(1/\delta))$ activations w.p. $\geq 1 - \delta$, consistent with Eq. 9.

Table 7: **Effect of pruning threshold $\alpha$ at $\varepsilon=3$ on the Document Classification task.** Numbers are TRR, ERR, and Avg. Per-schedule best operating point is bolded.

| $\alpha$ | Comprehensive | | | Progressive | | | Dynamic | | |
|---|---|---|---|---|---|---|---|---|---|
| | TRR | ERR | Avg. | TRR | ERR | Avg. | TRR | ERR | Avg. |
| 10 | .82 | .99 | .91 | .82 | .92 | .87 | .94 | .87 | .91 |
| 15 | .83 | 1.00 | .92 | .82 | .95 | .89 | .90 | .84 | .87 |
| **20** | **.85** | **1.00** | **.93** | .82 | .95 | .89 | **.87** | **.94** | **.92** |
| 25 | .83 | 1.00 | .92 | **.82** | **.95** | **.89** | .90 | .84 | .87 |
| 30 | .82 | .99 | .91 | .82 | .95 | .89 | .90 | .92 | .91 |
| 35 | .82 | 1.00 | .91 | .82 | .92 | .87 | .89 | .93 | .91 |
| 40 | .82 | .99 | .91 | .82 | .95 | .89 | .88 | .93 | .91 |
| 50 | .82 | .99 | .91 | .82 | .95 | .89 | .84 | .99 | .92 |
| 100 | .82 | .99 | .91 | .82 | .95 | .89 | .84 | .99 | .92 |
| 200 | .82 | .99 | .91 | .82 | .95 | .89 | .82 | .99 | .91 |

## B.5 Efficient Online Computation of rad

---

**Algorithm 2** Online computation of $\mathrm{rad}_k(t, \delta)$ via numerical inversion

---

**Inputs:** activations $n$, bad events $S$, tolerance $p^\star$, risk $\delta$
1: $\hat{p} \leftarrow S/n$; search interval $[0, \min(1 - p^\star, 1)]$
2: **for** binary search until numerical tolerance $\varepsilon_{\mathrm{num}}$ **do**
3:      midpoint $r$; set $u \leftarrow p^\star + r$
4:      compute $\mathcal{E}_t(u)$ (mixture integral; discretization acceptable)
5:      **if** $\mathcal{E}_t(u) \le 1/\delta$ **then**
6:          move left
7:      **else**
8:          move right
9: **return** $\mathrm{rad} \leftarrow r$

---

# C Full Sweeps Results

## C.1 Effect of Different Pruning Threshold $\alpha$

Table 7. We fix $\varepsilon=3$ and sweep the pruning threshold $\alpha \in \{10, 15, 20, 25, 30, 35, 40, 50, 100, 200\}$ under three schedules—Comprehensive (C), Progressive (P), and Dynamic (D). We evaluate the Document Classification task and report TRR, ERR, and their average; bold indicates the per-schedule best operating point.

## C.2 Effect of Different Initial $\varepsilon$

Table 8. With $\alpha=20$ fixed, we vary $\varepsilon \in \{0.001, 0.01, 0.1, 1.0, 3.0, 10.0, 100.0, 1000.0\}$ for three schedules on the same task. We report ERR, TRR, and ERR-Total; "–" denotes configurations that were not run.

# D Bounded Recycling Queue—Proofs and Engineering

This appendix supplies the details referenced as "bounded queue yields $O(1)$ amortized maintenance" and complements Eqs. 12–13.

## D.1 Deriving Equation No.13

This is the mathematical proof for Eq. 13. Let $\mathcal{E}_T$ be the set of distinct edited instances up to time $T$, and $r_j^t$ the cumulative number of retries of instance $j$ by time $t$ (Eq. 12), with $r_j^t \le R_{\max}$. Then

Table 8: **Epsilon ablation results** (SCOTUS Document Classification; $\alpha=20$).

| Method | Epsilon | ERR | TRR | Avg |
|---|---|---|---|---|
| **Comprehensive** | 0.001 | .22 | .97 | .60 |
| | 0.01 | .47 | .97 | .72 |
| | 0.1 | 1.00 | .97 | .99 |
| | 1.0 | 1.00 | .97 | .99 |
| | 3.0 | 1.00 | .85 | .93 |
| | 10.0 | .99 | .82 | .91 |
| | 100.0 | .99 | .82 | .91 |
| | 1000.0 | .99 | .82 | .91 |
| **Progressive** | 0.001 | .21 | .97 | .59 |
| | 0.01 | .37 | .97 | .67 |
| | 0.1 | 1.00 | .97 | .99 |
| | 1.0 | 1.00 | .97 | .99 |
| | 3.0 | .94 | .83 | .89 |
| | 10.0 | .94 | .83 | .89 |
| | 100.0 | – | – | – |
| | 1000.0 | – | – | – |
| **Dynamic** | 0.001 | .22 | .97 | .60 |
| | 0.01 | .47 | .97 | .72 |
| | 0.1 | 1.00 | .97 | .99 |
| | 1.0 | 1.00 | .97 | .99 |
| | 3.0 | .87 | .94 | .91 |
| | 10.0 | .20 | .98 | .59 |
| | 100.0 | .17 | .99 | .58 |
| | 1000.0 | .17 | .99 | .58 |

$$\sum_{j \in \mathcal{E}_T} r_j^T \ \leq \ R_{\max} \, |\mathcal{E}_T|.$$

Let $N_{\text{pruned}}(t)$ be the number of keys pruned by $t$, and $N_{\text{pending}}(t)$ the number of currently failing items waiting for a retry check. The queue length satisfies

$$|\mathcal{Q}_t| \ \leq \ R_{\max} \, N_{\text{pruned}}(t) + N_{\text{pending}}(t),$$

yielding $O(1)$ amortized time per maintenance trigger.

### D.2 IMPLEMENTATION NOTES

- **Queue policy**: FIFO with optional source dedup to prevent blow-ups; add age-weighting to avoid starvation.
- **Minimal GRACE Hartvigsen et al. (2023) edit**: prioritize add→expand→split to restore correctness with minimal perturbation.
- **Idempotence**: if the dequeued item is already fixed by other updates, drop it without consuming the retry budget.

## E  DATA AND MODEL

This appendix expands the experimental setup from Sec. 4.1.

### E.1  SCOTUS DOCUMENT CLASSIFICATION

- **Splits**: train (1946–1982), validation (rolling-window sampling), test (1982–1991), edit stream (1992–2009; each miss triggers an edit).
- **Model**: BERT-base ($\sim$120M)Breton et al. (2025).
- **Metrics**: TRR on the test set, ERR on the cumulative edit set. TRR measures the retention rate on test data, while the ERR measures the retention rate on edit data.

### E.2 ZSRE AND NQ QUESTION ANSWERING

- **Models**: T5-small ($\sim$60M) Roberts et al. (2020).
- **Decoding**: beam or top-$p$; post-processing (case/punctuation normalization).
- **Scoring**: F1/EM Hartvigsen et al. (2023) mapped to $s(x, y; f) \in [0, 1]$; adequacy threshold $a{=}0.9$ (as in the main text).

## F COMPLEXITY, IMPLEMENTATION, AND EXTENDED PSEUDOCODE

This appendix elaborates on implementation complexity and extends Algorithm 1 from the main text.

### F.1 TIME AND SPACE COMPLEXITY

- **Retrieval**: per layer $\ell$, nearest neighbor (Eqs. 1–2). Brute force $O(N_\ell d)$; with ANN (HNSW and IVF), expected $O(\log N_\ell)$ depending on recall.
- **Evidence logging**: for each fired key, compute the task score and update a constant-time counter; if recording counterfactual $\Delta_k$ (Eq. 4), add one suppressed forward (can share caches).
- **Maintenance**: per-trigger scanning of active keys and queue operations; amortized $O(1)$ (App. D).
- **Memory**: codebooks of size $\sum_\ell N_\ell$; per-key $C_k, B_k, (n, S)$ statistics; ANN index overhead is implementation-dependent.

### F.2 PLUGGABLE LOGGING AND ANYTIME GUARD

---

**Algorithm 3** Per-trigger evidence and guard (extends Algorithm 1)

---

**Inputs:** fired key $k$, sample $(x_t, y_t)$, threshold $a$, risk $\delta$
1: compute $s_t \leftarrow s(x_t, y_t; f_{\theta, \mathcal{C}})$; set $Z_{k,t} \leftarrow \mathbb{1}\{s_t < a\}$
2: update $n_k \leftarrow n_k + 1$, $S_k \leftarrow S_k + Z_{k,t}$, and accumulate $C_k, B_k$ per schedule
3: **if** counterfactual logging enabled **then**
4:     compute/cache $\Delta_k(x_t, y_t)$
5: **if** anytime guard enabled **then**
6:     compute $\mathrm{rad}_k(n_k, S_k, \delta)$ (Alg. 2 or closed-form bound)
7:     **if** $\hat{p}_k - p^\star > \mathrm{rad}_k$ **then**
8:         mark $k$ as prunable

---

### F.3 DYNAMIC TRIGGERING VIA e-PROCESS CHANGE DETECTION

Dynamic mode monitors $\widehat{\mathrm{TRR}}_t, \widehat{\mathrm{ERR}}_t$. If an e-process detector crosses an anytime threshold, a maintenance trigger fires; with safety margin $\gamma$ and bounded drift $V_T$, the expected number of triggers over horizon $T$ scales as $O(1+V_T/\gamma)$, matching the "avoids rapid toggling" claim in the main text.

### F.4 MULTI-KEY, MULTI-LAYER CONCURRENCY

If multiple layers fire on the same input, we use layer-wise independent accumulation with *deferred pruning*: an input only updates counters; actual pruning is decided at triggers, layer by layer, in parallel (with proper locking for index updates).

## G FORMULA ELABORATIONS

**Edited representation (Eq. 3) as gated routing.** Define a gate $g^{(\ell)}(x) \in \{0, 1\}$ based on $d^\star$ and $\varepsilon$,

$$\tilde{\boldsymbol{h}}^{(\ell)}(x) = g^{(\ell)}(x) \cdot \boldsymbol{v}_{i^\star}^{(\ell)} + \big(1 - g^{(\ell)}(x)\big) \cdot h_\theta^{(\ell)}(\boldsymbol{h}^{(\ell-1)}(x)).$$

**Metric-anchored vs. counterfactual harm (Eqs. 4, 5).** $\Delta_k$ quantifies loss *difference* from ablating $k$ on a specific example, while $H_k^{(a)}$ forms a *thresholded* indicator $Z_{k,t}$ for streaming decisions. They are complementary: metric-anchored harm yields a lightweight, anytime-testable statistic; optional $\Delta_k$ improves diagnostics (e.g., identifying neutral-but-redundant keys).

**Regret bound intuition (Eq. 11).** The policy is elimination-style: keys with $p_k > p^\star$ are removed within $O(\Delta^{-2} \log(1/\delta))$ activations (Theorem 2), while benign keys enjoy type-I control (Theorem 1). The overall expected regret collects a finite "exploration" term plus a $\widetilde{O}(\sqrt{T})$ stability term.

## H  REPRODUCIBILITY CHECKLIST AND RAW TABLES

### H.1  HARDWARE AND SOFTWARE ENVIRONMENT

Take conducting the SCOTUS-bert document classification with zsRE by **Dynamic (CleanEdit)** $\alpha$=20 as an example:

- GPUs: $1\times$ NVIDIA A100 PCIe 40GB.

- CPU: Intel Xeon Processor (Skylake, IBRS).

- RAM: 629GB system memory.

- Frameworks: PyTorch 2.4.1, Transformers 4.46.3, CUDA 12.4, Python 3.8.10.

- Random seeds: 42.

- Model: tomh/scotus-bert (BERT-based model with 120-million parameter).

- Editor: GRACE Hartvigsen et al. (2023).

- Pruning: CleanEdit: Dynamic schedule with $\alpha$, $\varepsilon = 3.0$.

- Training parameters: batch_size=1, edit_lr=1.0, n_iter=100, max_n_edits=1000.

- Experiment duration: Approximately 2.5 hours.

## I  LIMITATIONS AND FUTURE WORK

**Nonstationarity and drift.** The anytime guard controls false alarms, but fixed $p^\star$ and $a$ can become suboptimal under rapid drift. Adaptive thresholds and hierarchical guards are promising.

**ANN retrieval bias.** Imperfect recall in ANN can bias activation statistics. In practice, recheck high-frequency keys with exact NN to calibrate.

**Cross-layer interactions.** We accumulate evidence per layer; in tightly coupled models, multi-key interactions may require a joint (multivariate) guard.

**Beyond QA and Classification.** Extending $s(\cdot)$ to structured prediction or multimodal tasks may need sequence- or region-level adequacy thresholds.

## J    GLOSSARY OF TERMS AND SYMBOLS

Table 9: Key symbols

| Symbol | Meaning |
| --- | --- |
| $f_\theta$ | frozen backbone |
| $\mathcal{C}^{(\ell)}$ | layer-$\ell$ codebook (key/value/radius) |
| $\boldsymbol{k}_i^{(\ell)}, \boldsymbol{v}_i^{(\ell)}, \varepsilon_i^{(\ell)}$ | key $i$'s components |
| $i^\star, d^\star$ | nearest index and distance (Eqs. 1–2) |
| $\tilde{\boldsymbol{h}}^{(\ell)}$ | edited layer representation (Eq. 3) |
| $\Delta_k$ | counterfactual harm (Eq. 4) |
| $H_k^{(a)}, Z_{k,t}$ | metric-anchored harm and bad-event indicator (Eqs. 5–6) |
| $C_k, B_k$ | counters used by the pruning rule (Eq. 8) |
| $\alpha$ | pruning-count threshold |
| $p^\star, \delta$ | tolerated bad-event rate and risk for the guard |
| $\mathrm{rad}_k(t, \delta)$ | anytime radius (App. B) |
| $R_{\max}$ | retry cap for the queue (Eqs. 12–13) |
| TRR/ERR | test retention / edit retention (Sec. 4.1) |

**Cross-reference summary (ensuring main-text pointers are satisfied):**

- "Statistical reliability (CIs, paired tests, bootstrap) follows AppendixÃ" → App. A.
- "Optional anytime e-process guard" and related theorems → App. B.
- "Bounded recycling queue; $O(1)$ amortized maintenance" → App. D.
- "Full sweeps appear in AppendixC̃" → App. C.

**Optional further additions (placeholders reserved):**

1. *Multivariate guard for cross-layer keys*—extend App. B with joint e-processes.
2. *Key merge/dedup policies*—extend App. F with merge thresholds and centroid updates.
3. *Fine-grained error profiling*—extend App. C with confusion-style breakdowns (post-edit regressions, overgeneralization, domain drift).

## K    ETHICS STATEMENT

### K.1    THE USE OF LLM

In accordance with the ICLR 2026 policy on large language model (LLM) usage, we disclose that we used OpenAI GPT-4o for writing polish. The model was prompted with "Help me polish this paragraph, using a book-style expression." The LLM was only employed for language refinement and did not generate any novel content, experimental results, or analysis. All outputs were reviewed and verified by the authors, who take full responsibility for the final manuscript.

### K.2    LIMITATIONS AND SOCIAL IMPACT

CleanEdit maintains a memory of edits that may encode user-provided corrections. Care must be taken when edits contain sensitive text. Memory growth is actively controlled, but deployment should still budget for storage and retrieval. When the environment drifts adversarially, Dynamic triggers may fire frequently; the retry cap prevents oscillation, yet sustained drift may require retuning thresholds.

