# OpenReview forum: "CleanEdit: Retention-Aware Pruning and Bounded Replay for Lifelong Model Editing"
_ICLR.cc/2026/Conference — ICLR 2026 Conference Withdrawn Submission_

### Official Review · Reviewer_ZWd2 · 2025-10-29

**Soundness:** 3
**Presentation:** 2
**Contribution:** 2
**Rating:** 4
**Confidence:** 4

**Summary:**

The paper proposes a self-maintaining framework that enables language models to be continuously edited without degrading over time. It addresses the problem of edit memory pollution—where accumulated edits cause instability—by introducing two mechanisms: Retention-Aware Pruning, which statistically identifies and removes harmful or redundant edits, and Bounded Replay, which reuses supervision from pruned edits to prevent forgetting. Through these components and adaptive maintenance schedules, CleanEdit achieves more stable long-term performance, significantly improving the balance between retaining old knowledge and integrating new information compared to existing editors like GRACE and MEND.

**Strengths:**

* Well-motivated and timely problem: The paper tackles a crucial limitation in lifelong model editing—instability from unbounded memory accumulation—and reframes it as a memory curation problem, which is both novel and practically significant.

* Sound methodological design: The integration of counterfactual harm estimation, sequential testing, and bounded replay forms a coherent, self-maintaining system. The theoretical analysis (type-I control, sample complexity) strengthens the technical rigor of the work.

* Practical relevance and flexibility: The proposed scheduling strategies make the framework adaptable to different deployment contexts, from high-stability to high-throughput environments.

**Weaknesses:**

* Limited task diversity: The experiments mainly focus on medium-scale datasets (SCOTUS, zsRE). It remains unclear whether CleanEdit scales effectively to multi-hop tasks [1].

* Limited Baselines: The baseline methods the author compares are all from 23 years ago, but many related methods [2, 3] have emerged recently.

* Innovation boundaries could be clearer: Although the proposed pruning–replay loop is elegant, it builds upon existing ideas (memory-based editing + pruning + replay). The authors could better articulate what aspects are fundamentally new versus refined from prior work.


$$Ref:$$

[1] MQuAKE: Assessing Knowledge Editing in Language Models via Multi-Hop Questions, 2024.

[2] Reasons and Solutions for the Decline in Model Performance after Editing, 2024.

[3] O-Edit: Orthogonal Subspace Editing for Language Model Sequential Editing, 2024.

**Questions:**

* I aim to further investigate the performance degradation of CleanEdit on models, such as its impact on downstream tasks unrelated to editing, and how the norm of the model's editing parameters changes with the number of edits.

* It is better to consider adding a runtime/memory comparison table with GRACE, ROME, and MEND.

---

### Official Review · Reviewer_d2d4 · 2025-10-31

**Soundness:** 2
**Presentation:** 2
**Contribution:** 2
**Rating:** 2
**Confidence:** 4

**Summary:**

This paper proposes CleanEdit, a self-maintaining lifelong model-editing framework that actively curates edit memory to prevent accumulation-induced degradation. It introduces retention-aware pruning via counterfactual and metric-anchored harm estimates with anytime guarantees, plus bounded replay of pruned samples to avert forgetting. It also adopts three scheduling modes—Comprehensive, Progressive, and Dynamic to balance stability, efficiency, and deployment constraints.

**Strengths:**

1. The paper is well structured.
2. The mathematical proof seems unambiguous.

**Weaknesses:**

1. The method is proposed for sequential editing, but there is no scaling-up experiment to demonstrate the effectiveness of this method for dealing with a relatively large number of edits.
2. There is no side effect evaluation for the posted-edited model, which is crucial for sequential editing.
3. I think the authors should spend more on Figure 1 to make it easier for the reader. Maybe add more captions.
4. Lots of important baselines are missed in the experiment—for example, MEMIT, IKE, etc. Although MEMIT is not designed for sequential editing, it can perform multiple edits in a batch. Therefore, the author can compare the batch-editing method with their proposed method by fixing the number of edits.
5. The experiment datasets are too elementary. More complex editing datasets should be used to demonstrate the capability of CleanEdit to tackle complex scenarios. For example, the multihop editing dataset Mquake, etc

**Questions:**

1. If a specific edit is deemed detrimental, how does CleanEdit deal with it? Directly pruning it is likely not a good choice, because the information could be recalled in the future any time. More clarification is needed.

---

### Official Review · Reviewer_JpAB · 2025-11-01

**Soundness:** 2
**Presentation:** 3
**Contribution:** 2
**Rating:** 2
**Confidence:** 5

**Summary:**

The authors introduce CleanEdit for lifelong model editing which doesn’t suffer from performance decay through active memory management. The method includes a loop that supports maintenance of the memory with pruning via counterfactual harm estimation, and replay of pruned samples to prevent knowledge loss. The authors test this using a 60M model on two knowledge editing datasets.

**Strengths:**

1. Well motivated and relevant problem facing both researchers and practitioners in ML.
2. The technical approach is principled and novel addition to Grace.
3. Uses proper evaluation metrics with TRR and ERR.
5. Decent presentation and generally easy to follow.

**Weaknesses:**

1. The model is only 60M parameters, which makes it small in modern context and lacks many of the capabilities and properties of modern language models. This makes generalisation to real-world applications hard to gauge. Modern editing methods are typically evaluated on models that are >1B.
2. Some concepts are not explained fully, for example the time/dataset “blocks” are not clarified. Similarly the τ_TRR and τ_ERR don’t seem to be defined for the experiment either. These really need to be defined more explicitly and clearly.
3. The performance gap between the different schedules is very small (0.89 vs 0.91 vs 0.93) so it is not clear if they really matter. Further, these seem to be from single runs, which brings me to the next point.
4. The results are lacking proper statistical analyses. There appears to be just one run per experiment, meaning distributions, variance and effect sizes become very hard to gauge. This is a particular problem since the effects are very small in many cases, meaning they struggle to back up many of the performance claims. Without this, I think the claims of SOTA performance and stability/reliability are not fully backed up.
5. The appendix mentions 5 seeds with averaging and that 95 % CI intervals are presented. However, I do not find them anywhere. This is critical inconsistency and needs to be addressed.
6. I would have liked to have seen comparisons with other memory management baselines from continual learning like gradient-based sample selection.
7. There is a lack of discussion around the real world implications of TRR and ERR and their trade-offs. Do different applications/areas have the same weighting? Without this it’s harder to interpret the work and harder for practitioners to adapt this method to their application.
8. There are some key recent 2024/2025 related works are not discussed. These include: A) Composable Interventions (ICLR 2025) that studies how multiple interventions (including knowledge editing) compose and interact when applied sequentially to the same model, highly related experiments to the scenario CleanEdit addresses. B) WISE (NeurIPS 2024) another lifelong model editing method with a dual memory scheme (main memory + side memory) C) MEMOIR (2025) D) Memory reply methods like CORE (2024) and AMR.  These should be added to the related works section.
9. In table 5, ε=1 performs best but in table 1 it seems ε is set to 3 resulting in lower metrics on SCOTUS, why is this setting used?
10. There is limited analysis and discussion on the practical application of the method. What is the growth rate of the codebook under expected real-world circumstances? How does that compare with Grace? The time and space complexity overview in the appendix is very brief.
11. There are no general performance checks to verify that the model hasn't collapsed or lost significant general performance. I would have expected some general language benchmark like MMLU to quantify the effect of the method on general model performance, especially as edits build up.

### Minor
- \citep instead of \citet or \cite to get right parentheses.
- Some statements not backed up by references, e.g. L45 "While pruning and replay strengthen the intrinsic stability of edit memory"
- Something strange at start of L290.
- Many of the references are mangled. E.g. 505-510, 517-521, 556-561 and others. They seem to be some kind of Frankenstein version of two unrelated papers.

**Questions:**

1. How should ε be set initially? is it learned? Fixed? Per-key or global? Some guidance for practitioners or implementers would be helpful.
2. Is ε the same for all keys? The notation is ε_i^(l) which suggests it’s per key per layer, but then the experiments suggest global ε.
3.  How are time-blocks defined? What scale are they on? Years? Days?
4.  What is the justification for the 0.9 and 1.0 thresholds for a? Is it selected on validation through a sweep?
5.  It would be good if the authors could provide plots showing codebook size over time for CleanEdit vs GRACE
6.  Has the method been tested on models >1B parameters? >10B? Do findings generalise?
7.  What types of edits get pruned? Are they genuinely harmful or just hard examples?
8.  Is the e-process guard actually beneficial? I would have expected an ablation experiment here. The authors say this is optional, but it’s not clear when/why it is needed.
9.  (Note some questions also appear in the weaknesses section)

---

### Note · Authors · 2026-01-11

I have read and agree with the venue's withdrawal policy on behalf of myself and my co-authors.